# Embedding game: dimensionality reduction as a two player zero-sum game

## Abstract

Dimensionality reduction is often formulated as a minimization containing a sparse sum of attractive interactions and a dense sum of repulsive interactions $\sum_{ij} f(\|\mathbf{y}_i - \mathbf{y}_j\|)$ between embedding vectors. This dense sum is usually subsampled to avoid computing all $N^2$ terms. In this paper we provide a novel approximation to the repulsive sum by deriving a landmark-based lower bound and then maximizing this lower bound with respect to the landmarks. After inserting this approximation into the original objective we are left with a minimax problem where the embedding vectors minimize the objective by pulling on their neighbors and running away from the landmarks while the landmarks maximize the objective by pulling on the embedding vectors and running away from other nearby landmarks. We use gradient descent ascent to find saddle points and show that our method can produce high quality visualizations without ever explicitly computing any pairwise repulsion between embedding vectors.

## 1 Introduction

Dimensionality reduction algorithms can be useful in a wide variety of contexts. Reducing the dimensionality of vectors can reduce the computational burden on downstream tasks such as recognition or neighborhood searches. Reducing the dimensionality of inputs can also be a method to filter out unwanted variability in the original inputs. In the extreme case of reduction to 2 or 3 dimensions, it can be used to produce visualization of the input [14]. This is the case we will be concerned with in this paper.

Recent algorithms have yielded very impressive looking visualizations of complicated datasets [13, 8, 12, 11, 1]. Common to each of these algorithms is an objective function containing non-linear interactions between all (or nearly all) pairs of embedding vectors. There are a variety of approximations that have been proposed to approximate these all-pairs interactions. t-distributed Stochastic Neighbor Embedding (T-SNE) has taken inspiration from physical simulation and used the Barnes-Hut algorithm to cleverly discretize embedding space in a manner that allows for efficient approximation of all-pairs nonlinear interactions [13]. LargeVis and UMAP both use weighted edge sampling. At each iteration (or after several iterations), a random subset of interactions are chosen (with higher weighted interctions more likely to be chosen) and the subsampled objective is instead optimized [8, 12]. PyMDE uses a similar idea, but instead samples negative edges uniformly at random and these edges are fixed throughout training [1]. The tractable Latent Variable Model used landmarks to approximate the repulsion and relied on a sophisticated coarse graining scheme to reduce the number of pairwise interactions [11].

In this paper we will use a landmark approach for approximating nonlinear all-pairs interactions. We will derive a lower bound to a sum of all-pairs interactions which we then maximize with respect to the landmarks. We sill show that this requires much fewer landmarks than if we simply randomly sampled some fixed set of landmarks. Our algorithm can be interpreted as a two player game where

Submitted to 36th Conference on Neural Information Processing Systems (NeurIPS 2022). Do not distribute.

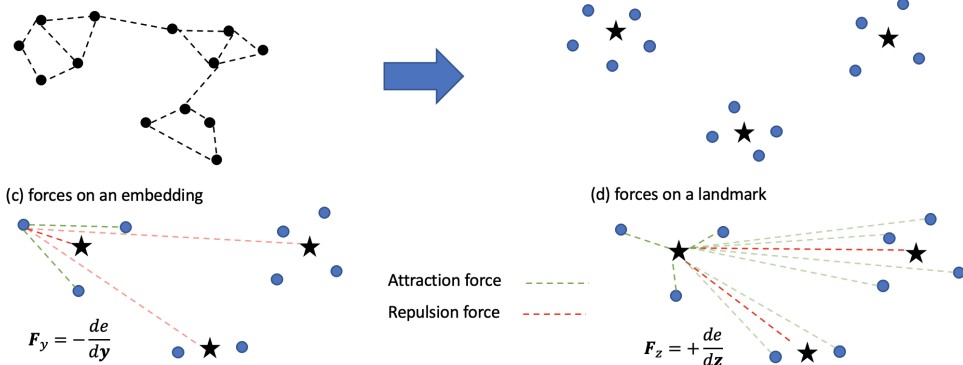

(a) compute K nearest neighbors for high-dim inputs

(b) learn low-dim embeddings (●) and landmarks (★) with gradient descent ascent

(c) forces on an embedding

(d) forces on a landmark

Attraction force

Repulsion force

$$\mathbf{F}_y = -\frac{de}{d\mathbf{y}}$$

$$\mathbf{F}_z = +\frac{de}{d\mathbf{z}}$$

Figure 1: Diagram of our method. Each embedding is attracted to the neighbors we compute in step a and repelled by all landmarks. There is no direct embedding-embedding repulsion. Landmarks are attracted to all embedding vectors and repelled from all other embedding vectors. These forces are derived in Eqs. 4, 5 as the gradients of the objective in Eq. 3.

the embedding vectors minimize the objective by pulling on their neighbors and running away from the landmarks while the landmarks maximize the objective by pulling on the embedding vectors and running away from other nearby landmarks. We use gradient descent ascent and show that our method can produce high quality visualizations.

## 2   Embedding Game

Our starting point closely follows [1] on Minimum Distortion embedding vectors (MDE). We assume we have a set of high-dimensional inputs $\{\mathbf{x}_i\}_{i=1}^N$ and we wish to produce a corresponding set of low-dimensional *embedding vectors* $\{\mathbf{y}_i\}_{i=1}^N$ which reveal interesting structure in the original inputs. To accomplish this we compute a sparse set of neighborhood edges: $\mathcal{N} = \{(i, j) : j \text{ is a k-nearest neighbor of } i\}$. We define two functions: $f(d)$ will penalize large distances between embedding vectors and $g(d)$ will penalize small distances between embedding vectors. We wish to find embedding vectors $\mathbf{y}$ which minimize:

$$\min_{\mathbf{y}} \sum_{i,j \sim \mathcal{N}} f(\|\mathbf{y}_i - \mathbf{y}_j\|) + \sum_{i,j} g(\|\mathbf{y}_i - \mathbf{y}_j\|) \tag{1}$$

Like many dimensionality reduction objectives, this contains a sum over $N^2$ terms. One method to avoid this unwieldy sum is simply to sample edges [8, 12, 1]. We proceed in a different fashion by defining a set of *landmarks* $\{\mathbf{z}_a\}_{a=1}^L$ and making a landmark-based approximation to the sum. Perhaps the simplest landmark-based approximation is simply to set the landmarks to be randomly chosen samples $\mathbf{z}_1 = \mathbf{y}_{i_1}, \mathbf{z}_2 = \mathbf{y}_{i_2}, \dots$ and approximate the all-pairs sum with $\frac{N}{L} \sum_{ia} g(\|\mathbf{y}_i - \mathbf{z}_a\|)$. This can actually be interpreted as another method of sampling edges. However we show in the Experiments section that this idea has several problems when using a small number ($< 300$) of landmarks.

Fortunately we can make a more powerful landmark-based approximation if $g(\|\mathbf{y}_i - \mathbf{y}_j\|)$ defines a positive semi-definite kernel function, in other words if the $N \times N$ matrix of pairwise interactions defined by $g$ is positive semi-definite for all $\{\mathbf{y}_i\}$. Some examples of $g$ satisfying this property are $g(d) = \exp(-d^2)$ and $g(d) = 1/(1 + d^2)$. Unfortunately our bound will not work with $g(d)$ that go to infinity as $d \to 0$ which are sometimes used in the literature. However we'll show that we can learn high quality embedding vectors without this "infinite as $d$ goes to zero" property.

**Key inequality**: if $g(\|\mathbf{y}_i - \mathbf{y}_j\|)$ is a positive semi-definite kernel function, then for any $\{\mathbf{y}_i\}_{i=1}^N$ and $\{\mathbf{z}_a\}_{a=1}^L$:

$$\left[ \sum_{i,a} g(\|\mathbf{y}_i - \mathbf{z}_a\|) \right]^2 \leq \left[ \sum_{ij} g(\|\mathbf{y}_i - \mathbf{y}_j\|) \right] \left[ \sum_{a,b} g(\|\mathbf{z}_a - \mathbf{z}_b\|) \right] \tag{2}$$

66  We are unaware of this inequality being presented in the literature but we prove it in the Appendix.
67  The key to our proof is to replace $g(\|\mathbf{y}_i - \mathbf{y}_j\|)$ with the inner product between high dimensional
68  vectors $\phi_i \cdot \phi_j$. This is allowed because of our assumption that $g$ defines a kernel function. A useful
69  property of our inequality is that with at most $N$ landmarks, there exist $\mathbf{z}$ which yield *equality*, rather
70  than inequality. This can be seen by simply setting the $\mathbf{z}_a = \mathbf{y}_i$ when $L = N$. So long as we
71  have enough landmarks, our approximation should yield the same result as the original "all-pairs"
72  repulsion objective. Of course we hope that our approximation is useful for $L \ll N$.

73  To generate our landmark-based approximation to the all-pairs sum in Eq. 1 we divide both sides of
74  Eq. 2 by $\sum_{ab} g_{ab}$ and then maximize $\sum_{ia}^2 / \sum_{ab}$ with respect to $\mathbf{z}$. This yields the tightest lower
75  bound to the sum $\sum_{ij} g(\|\mathbf{y}_i - \mathbf{y}_j\|)$. We then replace the pairwise sum in Eq. 1 with our tightest
76  lower bound to yield the minimax problem we ultimately try to solve:

$$\min_{\mathbf{y}} \ \max_{\mathbf{z}} \ \sum_{i,j \sim \mathcal{N}} f(\|\mathbf{y}_i - \mathbf{y}_j\|) + \frac{\left[ \sum_{i,a} g(\|\mathbf{y}_i - \mathbf{z}_a\|) \right]^2}{\sum_{a,b} g(\|\mathbf{z}_a - \mathbf{z}_b\|)} \tag{3}$$

77  The denominator in the right-hand term contains $LN$ interactions (between all landmark-embedding
78  vector pairs) and the numerator contains $L^2$ interactions (between all landmark-landmark pairs).

## 2.1  Gradient descent ascent (GDA)-based optimization

```
# x:   input vectors (shape:  (n,m))
# k,l:  num neighbors, num landmarks
# f,g:  attractive, repulsive penalty functions

edges = knn_edges(x,k) # k nearest neighbors for each input
y = laplacian_eigenmap(edges) # init embedding vectors
z = sample_landmarks(y,l) # init landmarks as randomly chosen embedding
 vectors

for i in range(n_iter):
    # pairwise distances
    yy = norm((y[edges[0]] - y[edges[1]],dim=1) # shape:  (n*k)
    yz = cdist(y,z) # shape:  (n,l)
    zz = cdist(z,z) # shape:  (l,l)

    # energy
    e = f(yy).sum() + (g(yz)**2).sum() / g(zz).sum() # Eq.  3

    # gradients
    e.backward()

    # updates
    y -= eta_y / (y.grad**2).mean().sqrt() * y.grad
    z += eta_z / (z.grad**2).mean().sqrt() * z.grad
```
**Algorithm 1:** PyTorch-style pseudocode for embedding game

80  We provide PyTorch-style pseudocode in Alg. 1. We use a rescaled gradient descent-ascent algorithm
81  to find saddle points of Eq. 3. This rescaling is helpful as the embedding gradients are much smaller
82  than the landmark gradients and the rescaling lets us set $\eta_y, \eta_z$ to be similar magnitudes. In principle
83  this rescaling could lead to convergence issues, however this was not problem in practice.

84  As is standard practice [8, 1, 11], we initialize the embedding vectors using Laplacian eigenmaps [3].
85  This method sets $\mathbf{y}$ to be the 2nd and 3rd smallest eigenvectors of the normalized graph Laplacian
86  defined by the k-nearest neighbor graph. The $\mathbf{y}$ are rescaled so each dimension is unit variance.
87  In practice we don't find exact eigenvectors but rather 100 power iterations to approximate these
88  eigenvectors. We initialize the landmarks by randomly sampling embedding vectors after laplacian
89  initialization.

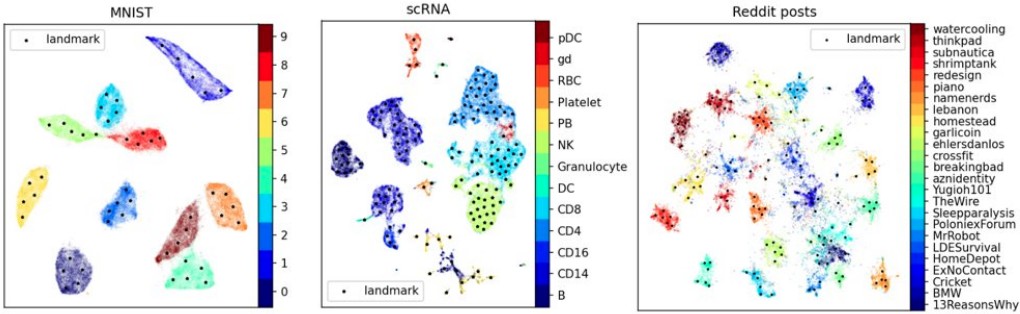

Figure 2: embedding vectors (color) and landmarks (black) generated by our algorithm on three datasets.

Both in theory and in practice, choosing the learning rates for GDA problems can be much more complicated than for simple gradient descent problems. In the experiments section we explore how various learning rate choices impact convergence.

## 2.2 Force-based interpretation of the game

It is useful to consider the forces on each player of the game. The force on each embedding takes the form:

$$\mathbf{f}_i = -\frac{dE}{d\mathbf{y}_i} = \alpha_1 \sum_{j \sim \mathcal{N}_i} f'_{ij} \frac{\mathbf{y}_j - \mathbf{y}_i}{\|\mathbf{y}_j - \mathbf{y}_i\|} + \alpha_2 \sum_a g'_{ia} \frac{\mathbf{y}_i - \mathbf{z}_a}{\|\mathbf{y}_i - \mathbf{z}_a\|} \tag{4}$$

where $\alpha_1, \alpha_2$ are non-negative constants. The force has two sources: each embedding feels attraction to its neighbors and repulsion from all landmarks. In practice, $g'$ will decay with distance so the repulsion is strongest from nearby landmarks. In simple terms *the embedding vectors run towards their neighbors and away from the landmarks.*

Because the landmarks are maximing the objective, the force is now the positive gradient. This takes the form:

$$\mathbf{f}_a = +\frac{dE}{d\mathbf{z}_a} = \beta_1 \sum_i g'_{ia} \frac{\mathbf{y}_i - \mathbf{z}_a}{\|\mathbf{y}_i - \mathbf{z}_a\|} + \beta_2 \sum_b \frac{\mathbf{z}_a - \mathbf{z}_b}{\|\mathbf{z}_a - \mathbf{z}_b\|} \tag{5}$$

where $\beta, \beta_2$ are non-negative constants. This force also has two source. Landmarks are attracted to the embedding vectors (but prefer close embedding vectors) and repelled by other landmarks. In simple terms *the landmarks run towards the closest embedding vectors and away from the other nearby landmarks.*

# 3 Visualization results

We show visualizations produced by our algorithm on three different datasets (i) the classic MNIST dataset (ii) a single cell RNA dataset (iii) 25k reddit posts from 25 different subreddits. The resulting embedding vectors and landmarks are shown in Fig. 2. The penalty functions we use are:

$$f(d) = \log(1 + d^2) \qquad g(d) = \frac{\lambda}{1 + d^2} \tag{6}$$

$\lambda$ is a hyperparameter which we tune for each dataset. The "log 1-plus" attractive penalty has seen successes in previous works so we stick with it [8, 1]. Intuitively this function penalizes large distances less than than a more intuitive $d^2$ penalty, which may be important in the presence of noisy neighborhood graphs. Our "cauchy" repulsive penalty $g$ is a more unusual choice. Unlike other works, $g$ does not approach infinity as the distance goes to zero. This is important as our bound does not work when $g(0) \to \infty$. However, this penalty still decays to zero as $d \to \infty$, meaning that repulsion is strongest between nearby vectors. We show empirically that we can generate high quality visualizations with this class of distortion function.

For every dataset we adopt the same learning rate schedule. We perform 3k GDA iterations with $\eta_y = 0.03, \eta_z = 0.3$. We divide both learning rates by 10 and perform 3k more iterations. There is

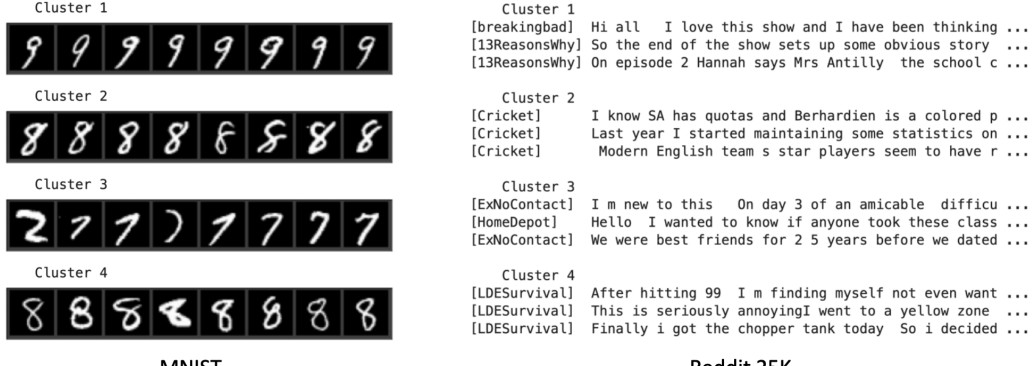

Figure 3: Clustering derived by assigning each embedding to the nearest landmark. We show random subsets of each cluster for the first four clusters in the MNIST dataset and the Reddit25K dataset

120 no stochasticity in our system as we are doing full batch updates, however the learning rate decay
121 still seems helpful for improving the resulting visualization.

### 3.1 Datasets

123 **MNIST** The classic MNIST dataset contains 70,000 grayscale images of size 28x28, each containing
124 a single handwritten digit. The generated 2D visualization is shown for $K = 15$ neighbors, $L = 50$
125 landmarks and $\lambda = 0.001$.

126 **scRNA** The scRNA dataset contains 40,000 PCA embedding vectors of single cell mRNA transcrip-
127 tomes from COVID-19 patients. The input vectors are 30 dimensional. This dataset originates from
128 [15] and can be conveniently be download from [1]. We use $K = 15$, $L = 150$ and $\lambda = 0.01$.

129 **25K Reddit Posts** This dataset contains 25,000 reddit posts from 25 different subreddits (1000 posts
130 per subreddit). This is generated by sampling 25 subreddits from the larger 1 million post dataset
131 which can be downloaded from Kaggle [6]. To generate feature vector for each post, we use the
132 strategy detailed in [2] and create a weighted average of GloVe vectors used in the post. These 300
133 dimensional "post vectors" are then fed into our algorithm. We use $K = 15$, $L = 150$ and $\lambda = 0.01$.

### 3.2 Landmark-based clustering

135 In each case the landmarks appear to tile the high density regions of embedding space. This suggests
136 a method to cluster the data. Assign each embedding to the closest landmark in embedding space:

$$c_i = \operatorname*{argmin}_{a} \|\mathbf{y}_i - \mathbf{z}_a\| \tag{7}$$

137 One might expect similar results by running KMeans on the embedding vectors, but a difference is
138 that we already have the landmarks (cluster centers) as a result of running our algorithm. In Fig. 3
139 we show randomly chosen inputs that are assigned to selected landmarks for the MNIST and Reddit
140 post datasets. This a can provide an interesting way to quickly visualize data. Instead of directly
141 visualizing the embedding vectors, one can examine clusters derived from the learned landmarks.

## 4 Duality and learning rates

143 Choosing the learning rates for this problem is non-trivial because we have a non-convex non-concave
144 minimax optimization. However we provide a rule of thumb which is to choose the landmark learning
145 rate sufficiently large relative to the embedding learning rate. This is motivated by experiment and
146 extrapolation from theoretical results on nonconvex-concave optimization.

147 **Duality** In general there is a duality gap for our optimization in Eq. 3. In other words the order of the
148 optimization (min-max vs. max-min) is extremely important for this problem. In fact the reversed
149 "max-min" problem admits a completely degenerate set of landmarks. For any fixed set of landmarks

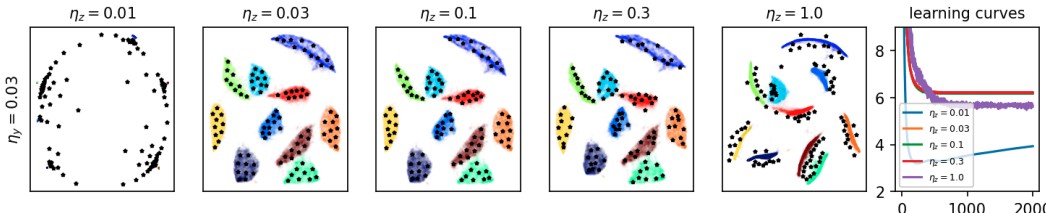

Figure 4: Varying the landmark learning rate $\eta_z$ with the embedding learning rate fixed at $\eta_y = 0.03$ (for the MNIST dataset). Too small, and the embedding vectors collapse to a few points ($\eta_z = 0.01$). Too large and the embedding vectors again appear to begin to collapse, although not as extreme here. There appears to be a window of learning rates where we observe nice clustering.

(z), the embedding vectors (y) can achieve 0 error by: one, setting the first sum in Eq. 3 to zero by collapsing to a point $\mathbf{y}_i = \mathbf{y}_j$ for all $i, j$, and two: setting the numerator of the second term in Eq. 3 to zero by running off to infinity. So for any fixed $\mathbf{z}$, we have $\min_y e(\cdot, \mathbf{z}) = 0$ and therefore $\max_z \min_y e = 0$. Ultimately we care about generating useful visualizations of data, so having all our embedding vectors collapsed at a single point, infinitely far away from the landmarks, is not good.

**Learning rates** Intuitively we should set the landmark learning rate to be relatively fast compared to the embedding learning rate. This way, the landmarks can approximately perform $\max_z e(\mathbf{y}, \cdot)$ before the embedding vectors have a chance to update appreciably. Then the embedding vectors can perform approximate gradient descent on the objective $\max_z e(\cdot, \mathbf{z})$ and the algorithm is more likely to find solutions to the original $\min\max$ problem.

Rigorously justifying this intuition, that setting $\eta_y \ll \eta_z$ will find a solution of $\min_y \max_z$, is rather challenging in the case of nonconvex-nonconcave objective like ours. In the simpler case of a nonconvex-concave problem, this intuition (choose a fast learning rate for the inside-maximization) can be rigorously shown to be correct [7]. We'll show via experiment that this intuition seems to be useful.

**Experiment** We run our algorithm on the MNIST dataset with $k = 15$, $l = 100$, and $\lambda = 0.001$. We'll fix $\eta_y = 0.03$ and vary $\eta_z$. Learning curves and visualizations after 2k iterations of Alg. 1 are shown in Fig. 4. Too small, and the embedding vectors collapse to a few points ($\eta_z = 0.01$). Too large and the embedding vectors again appear to begin to collapse, although not as extreme here. There appears to be a window of learning rates where we observe nice clustering. This seems to be explained by our intuitions on the duality problem.

When $\eta_z$ is too small, the embedding vectors can begin to minimize first and perform $\min_y e(\cdot, \mathbf{z})$, which we argued can be 0 when the embedding vectors collapse to a point and move away from the landmarks. When $\eta_z$ is too large, then again the landmarks are not performing the inner loop maximization, and the embedding vectors again can perform $\min_y e(\cdot, \mathbf{z})$, by collapsing to a point. In Fig. 4 we see the embedding vectors begin to collapse, although its not as extreme as when $\eta_z$ was too large.

## 5 Comparison with sample-based techniques

We'll compare to two fixed-sample-based methods for approximating the all-pairs sum in Eq. 1. The first method is used by [1] and simply samples $L$ edges for each node. The second method randomly designates $L$ embedding vectors at the start of training to be the landmarks and replaces the sum using the landmarks. This method is not widely used in practice, and as we'll see it leads to surprisingly low quality visualizations. We call these methods "fixed-sample" because these edges/landmarks are sampled once at the start of training and then fixed for all subsequent iterations.

We evaluate each method a) qualitatively by looking at the resulting visualizations and b) quantitatively by comparing the value of the "all-pairs" objective (Eq. 1) using the generated embedding vectors. We observe that the sampled-edge-based algorithm seems to outperform our algorithm for fixed $L$, while the sample-landmark-based algorithm dramatically underperforms.

**Sampled edges** At the start of training, we sample $L$ edges $(i, j)$ uniformly at random for each node $i$. We call this set of edges $\mathcal{N}^-$. The all-pairs sum $\sum_{ij}$ is then replaced with a sum over these edges and reweighted by a factor $N/L$. So we optimize the objective:

$$\min_{\mathbf{y}} \sum_{i,j \sim \mathcal{N}} f(\|\mathbf{y}_i - \mathbf{y}_j\|) + \frac{N}{L} \sum_{i,j \sim \mathcal{N}^-} g(\|\mathbf{y}_i - \mathbf{y}_j\|) \tag{8}$$

**Sampled landmarks** At the start of training, we designate $L$ embedding vectors uniformly at random to be landmarks. We call this set of embedding vectors $\mathcal{E}$. We again replace the all-pairs sum in Eq. 1 with a reweighted sum over landmarks:

$$\min_{\mathbf{y}} \sum_{i,j \sim \mathcal{N}} f(\|\mathbf{y}_i - \mathbf{y}_j\|) + \frac{N}{L} \sum_{i} \sum_{j \sim \mathcal{E}} g(\|\mathbf{y}_i - \mathbf{y}_j\|) \tag{9}$$

By defining the set of negative edges $\mathcal{N}^- = \{(i,j) : i = 1, 2, ..., N \text{ and } j \in \mathcal{E}\}$, we can rewrite Eq. 9 to look identical to Eq. 8. In other words the sampled landmark scheme can be interpreted as just another method to sample negative edges.

Averaging over edge or landmark choices, both schemes yield unbiased estimators $\langle \frac{N}{L} \sum_{ij \sim \mathcal{N}^-} \rangle = \sum_{ij}$. But because these are chosen once at the start of training and then fixed, these actually yield zero variance but biased estimates. In other words both methods generates a biased estimate of $\sum_{ij}$ and its gradients. If our method exactly computed the global maxima with respect to the landmarks for a fixed set of embedding vectors, it could as well be regarded as a zero-variance biased estimate of the all-pairs sum. However, it does not find the global maxima, so there is some variance in our estimates (due to for instance random initialization of the landmarks).

**Experiment training details** In all three experiments (sampled edges, sampled landmarks, and our method) we use the same hyperparameters. We train on the MNIST dataset. We use the $f, g$ described in Eq. 6. We set $\lambda = 0.001$. We use $K = 15$ for the neighborhood graph. We compare $L \in \{1, 3, 10, 30, 100, 300\}$. For the sampled edges and sampled landmarks experiment, we only have a learning rate for the embedding vectors. We use the same learning rate schedule for $\eta_y$ in both settings: we perform 2k iterations with $\eta_y = 0.03$, then 2k more with $\eta_y = 0.003$. For the experiment using our method, we have the additional parameter $\eta_z$ and we set it at $\eta_z = 10\eta_y$.

## 5.1 Visualization (qualitative comparison)

The resulting visualizations are shown in Fig. 5. As the number of landmarks $L$ increase beyond 30, both the optimized-landmark and sampled-edge algorithms yield nearly identical visualizations. This seems reasonable, as $L$ increases each algorithm yields a better and better estimate of the all-pairs objective so beyond a certain threshold of $L$, all algorithms should ultimately yield the same visualizations. For $L = 1, 10, 30$ the optimized landmark algorithm suffers from degeneracies not seen in the sampled-edge algorithm. In particular we observe a number of "clusters" appear to collapse to single points.

Intuitively when there are more clusters than landmarks, there is no mechanism for the optimized landmark algorithm to prevent some of the clusters from collapsing. In the landmark algorithms there is no direct embedding-embedding repulsive forces. If there is no landmark inside a cluster of embedding vectors which are attracting each other, there is no outward repulsion preventing this cluster from collapsing to a point as we see in the cases $L = 1, 3, 10$ where there are not very many landmarks. The sampled-edge method seems to provide a more robust mechanism for avoiding this embedding collapse.

The sampled-landmark algorithm gave nearly complete embedding collapse for all $L$ we tested (note that at some point, for sufficiently large L all three methods should yield the same result). All embedding vectors which were not designated as landmarks simply collapsed towards a single point while the landmark vectors were repelled to an exterior ring around the origin. It may seem surprising how different this result is from the sampled edges experiment, given that it can be interpreted as another way to generate negative edges. This result shows the importance of the exact method used to approximate the all-pairs repulsive sum.

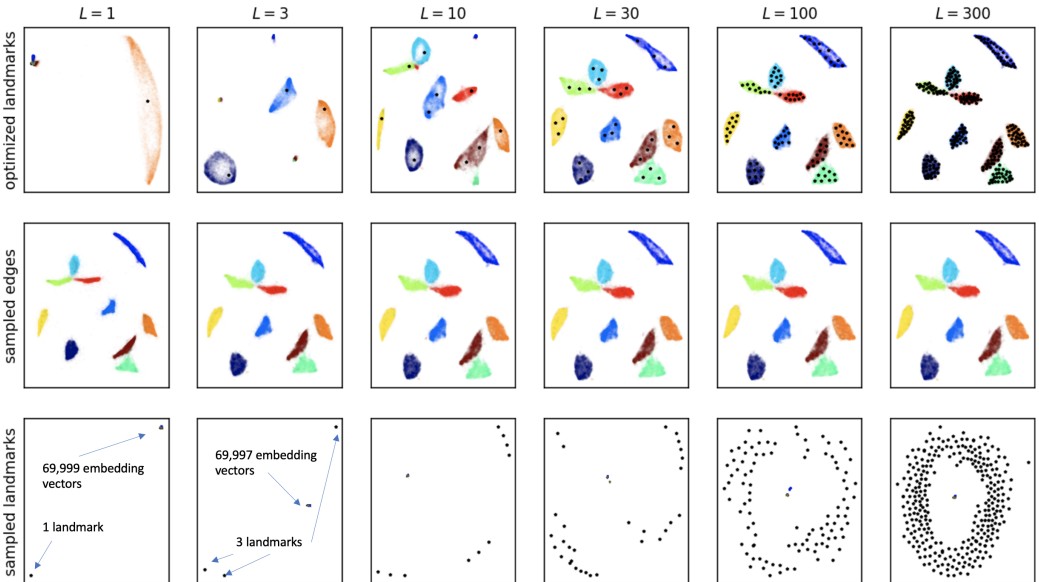

Figure 5: Comparing our optimized landmark method for approximating the all-pairs repulsion in Eq. 1 to simple edge-sampling and landmark-sampling methods. Black dots indicated landmarks (not relevant/present in sampled edges figures). The edge sampling method seems to outperform our method for a fixed L. When we don't have enough landmarks (L=1,3,10) we see clusters of embedding collapse to points. However our method is much better than a simple random sampling of landmarks. When we randomly sample and fix landmarks, nearly all the embedding vectors which were not designated as landmarks collapse to a single point, while the landmark vectors repel from each other and other embedding vectors.

## 5.2 Quantitative Results

We also compare the sampled-edge and the optimized-landmark algorithms quantitatively in Fig. 6. We don't show the sampled-landmark method as it is far worse than the sampled-edge or optimized-landmark methods. All curves in this Figure are for $L = 10$.

In (a) we show the energy we are actually optimizing (Eq. 3) for optimized-landmarks and Eq. 8 for sampled-edges). This indicates that both algorithms are at least optimizing the approximations, and the degeneracies we observed in Fig. 5 are not a failure of the optimization routine. In (b) we show an unbiased approximation to the "true" energy at each iteration (Eq. 1). This is done by randomly sampling a large number of edges from the all pairs sum, and these edges are chosen i.i.d. at each iteration

For both orange and blue, the energy we optimize is less than the all-pairs energy, and this difference can be regarded as analogous to a generalization gap. Each algorithm "overfits" to the energy we optimize, but this "overfitting" appears worse for our method than the sampled edge method.

We observe that the sampled-edge algorithms yields a lower all-pairs energy (the energy we truly wish to optimize). This is in agreement with the fact that the sampled edge method yields the highest quality visualizations with a small number of sampled edges (Fig. 5). In (c) we plot the root-mean-squared error between the gradients of the approximate energy and gradients of the estimated true energy. Our method produces a more faithful gradient estimate at nearly all times in training. This is extremely surprising as the final true energy, after many gradient updates is lower for the sampled edge method which gives a worse gradient estimate in terms of mean squared error. This result shows how the exact details used to approximate the sum can be very important.

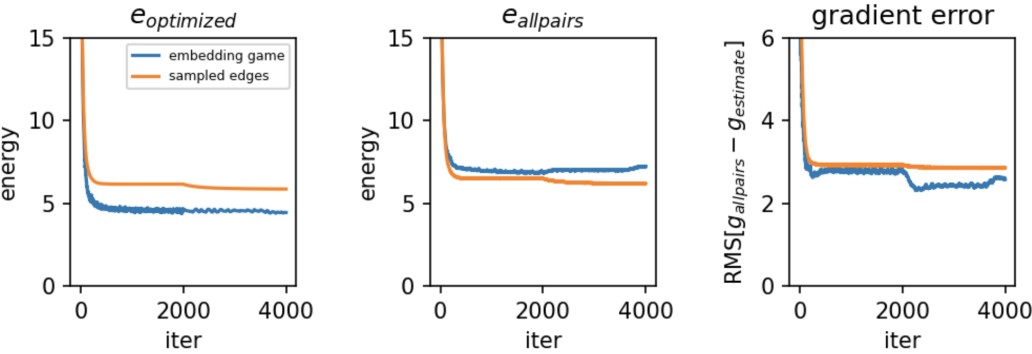

Figure 6: Quantitative comparison between our method (optimized landmarks) and sampled edge method. The left figure show the energy each method actually optimizes. The middle figure shows the "all-pairs" energy (Eq. 1).

## 6 Related Work

**Landmark methods** Instead of dealing with $N^2$ nonlinear pairwise interactions, landmark based approaches instead designate a small set of $n$ landmarks, and instead work a smaller set with $nN$ iterations. Unlike our method where individual landmarks do not correspond to any particular sample, most landmark approaches use sampling to designate certain samples as landmarks. For dimensionality reduction, landmark approaches have been applied ISOMAP by [4], and to stochastic neighbor embedding vectors by [10]. Finally more sophisticated landmark sampling schemes have been used by [11].

Perhaps the most well-known class of landmark methods is the Nyström method [16], which is used to approximate $N \times N$ kernel similarity matrices $g(\mathbf{x}_i, \mathbf{x}_j)$ with two smaller $N \times n$ and $n \times n$ matrices. Our method actually can be understood from the kernel perspective. The heart of our method is approximating the sum over all pairs of interactions $\sum_{ij} g(\mathbf{x}_i, \mathbf{x}_j)$ which we can interpret as the inner product of the vector of all ones and the kernel similarity matrix $\mathbf{1}^\top \mathbf{G} \mathbf{1}$.

**Game formulations of learning** This work falls into a category of works formulating well-established algorithms like multidimensional scaling, principle components analysis, and whitening as a game [9, 5]. This work is formulating a certain class of nonlinear embedding problems as a game.

## 7 Discussion

This paper presents a novel method for approximately the all-pairs repulsive term present in many manifold learning algorithms. When the nonlinear repulsion terms are described by a kernel function, we can derive a lower bound for the all-pairs sum which we then maximize to find the tightest lower bound. We show that this optimization requires much fewer landmarks than would be required if we instead just randomly designated embedding vectors to be landmarks. However, compared to sampling edges randomly this scheme still requires more computation to achieve comparable visualizations.

To make this method more useful, future work should find automated schemes for performing the minimax optimization, so a user does not have to specify learning rates. This might me much more challenging here than for a minimization problem because in general there is no guarantee that an increase or decrease in the objective means we are getting closer to a saddle point. In practice however, we observed success by setting larger learning rates for $\eta_z$ and smaller for $\eta_y$. This might suggest that finding a quick and robust inner loop maximization, with outer loop gradient steps could be a promising direction.

It would be interesting to apply our method to other regimes. In particular finding high dimensional embedding vectors. Additionally, exploring the behavior of this method in the online setting is promising, as our method for approximating a sum of all-pairs pairwise interactions does not actually require any pairwise distances between embedding vectors to be computed.

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

## Proof of Inequality in Eq. 2

We restate the claim first. If $g(\|\mathbf{y}_i - \mathbf{y}_j\|)$ is a positive semi-definite kernel function, then for any $\{\mathbf{y}_i\}_{i=1}^N$ and $\{\mathbf{z}_a\}_{a=1}^L$:

$$\left[\sum_{i,a} g(\|\mathbf{y}_i - \mathbf{z}_a\|)\right]^2 \leq \left[\sum_{ij} g(\|\mathbf{y}_i - \mathbf{y}_j\|)\right]\left[\sum_{a,b} g(\|\mathbf{z}_a - \mathbf{z}_b\|)\right] \qquad (10)$$

*Proof:* Because we have assumed that $g$ is a kernel, Mercer's theorem allows us to replace all the $g(\cdot)$ with inner products between high dimensional vectors. Specifically for any $\{\mathbf{y}_i\}_{i=1}^N$ and $\{\mathbf{z}_a\}_{a=1}^L$ there exist $\{\boldsymbol{\phi}_i\}_{i=1}^N$ and $\{\boldsymbol{\psi}_a\}_{a=1}^L$ such that

$$g(\|\mathbf{y}_i - \mathbf{z}_a\|) = \boldsymbol{\phi}_i \cdot \boldsymbol{\psi}_a \qquad g(\|\mathbf{y}_i - \mathbf{y}_j\|) = \boldsymbol{\phi}_i \cdot \boldsymbol{\phi}_j \qquad g(\|\mathbf{z}_a - \mathbf{z}_b\|) = \boldsymbol{\psi}_a \cdot \boldsymbol{\psi}_b \qquad (11)$$

Now proof of Eq. 10 is a simple matter of proving vector inequalities. First we define the sums $\bar{\boldsymbol{\phi}} := \sum_i \boldsymbol{\phi}_i$ and $\bar{\boldsymbol{\psi}} := \sum_a \boldsymbol{\psi}_a$. We can replace the sums of $g$ with our vectors. So the left hand side is

$$\left[\sum_{ia} g(\|\mathbf{y}_i - \mathbf{z}_a\|)\right]^2 = (\bar{\boldsymbol{\phi}} \cdot \bar{\boldsymbol{\psi}})^2 = \|\bar{\boldsymbol{\phi}}\|^2 \|\bar{\boldsymbol{\psi}}\|^2 Cos[\bar{\boldsymbol{\phi}}, \bar{\boldsymbol{\psi}}]^2 \qquad (12)$$

And the right hand side of Eq. 10 is:

$$\left[\sum_{ij} g(\|\mathbf{y}_i - \mathbf{y}_j\|)\right]\left[\sum_{a,b} g(\|\mathbf{z}_a - \mathbf{z}_b\|)\right] = \|\bar{\phi}\|^2 \|\bar{\psi}\|^2 \tag{13}$$

Because $Cos^2 \leq 1$, the left hand side is always less than the ride hand side so we have therefore proved Eq. 10.

