# OpenReview forum: "Embedding game: dimensionality reduction as a two-person zero-sum game"
_NeurIPS.cc/2022/Conference — NeurIPS 2022 Submitted_

### Official Review · Reviewer_aU48 · 2022-07-11

**Rating:** 4
**Confidence:** 4
**Soundness:** 3 good
**Presentation:** 3 good
**Contribution:** 2 fair

**Summary:**

- This paper presents a new approximation approach for dimensionality reduction formulation.
- Traditional dimensionality reduction is formulated as minimization of sum of attractive interaction and sum of repulsive interaction between embedding vectors. The proposed approach approximates the repulsive sum with maximizing the landmark-based lower bound w.r.t. landmarks to make it a minimax problem
- The proposed approach is optimized by a gradient descent ascent approach. And the selection of the hyperparameters are discussed. Experimental results showed the proposed approximation gives high quality clustering.

**Questions:**

- Do you have any mathematical analysis of the learning rate selection besides the experiments?
- Besides fixed iteration, maybe use energy difference as the termination condition of the loop?
- For Figure 6, energy of sampled edges are almost identical for e_optimized and e_allpairs which indicates sampled_edges has a better approximation than embedding_game. Why do you need RMS(gradients) other than RMS(energy) to tell the approximation quality?

**Limitations:**

Authors did mentioned the limitations and looking forward to seeing the approach to be applied to dimensionality increase problem.

**Strengths And Weaknesses:**

# Strengths
- Clever formulation of the approximation in section 2
- Clear interpretation of the math formulation, i.e. the embedding vectors run towards their neighbors and away from the landmarks....
- Clear visualization to show the clustering quality

# Weakness
- Quantitative results are not impressive, proposed method performs worse than sampled-edge algorithms when sampling size is the same. the proposed approach needs more computation to achieve comparable results.
- The discussion of the learning rate selection is not sufficient, neither with a mathematical analysis nor a more comprehensive combination (current work fix \eta_y, any other options?)

---

> ### Author Response · Authors · 2022-08-02
> **Reply to aU48**
>
> *Quantitative results are not impressive, proposed method performs worse than sampled-edge algorithms when sampling size is the same. the proposed approach needs more computation to achieve comparable results.*
>
> We agree. Our recommendation is still to use edge-based algorithms for this task. However, we still believe our findings are worthy of publication as they are useful for future researchers interested in this question of edge vs landmark-based approximations of the all pairs objective.
>
> *The discussion of the learning rate selection is not sufficient, neither with a mathematical analysis nor a more comprehensive combination (current work fix \eta_y, any other options?)*
>
> This is really a major stumbling point of nonconvex-nonconcave optimizations, that theoretical guarantees are hard to achieve. However, this does not mean that simple GDA can't be a useful optimization tool, and our experiments suggest that it can indeed be used to achieve useful optimizations. If anything, we believe this to be a positive contribution of the paper, as it provides on piece of evidence for future researchers that simple GDA on a nonconvex-nonconcave objective can indeed be extremely practically useful.
>
> Questions:
> *Do you have any mathematical analysis of the learning rate selection besides the experiments?*
>
> In section 4, in the **learning rates** section, we provide some discussion. In particular, setting the learning rate for the landmarks to be large is **not** an empirical finding. It is theoretically motivated: performing the max w.r.t. landmarks is meaningful for any set of embeddings. While performing the min w.r.t. embeddings with landmarks fixed is not meaningful as they will run off to infinity.
>
> *Besides fixed iteration, maybe use energy difference as the termination condition of the loop?*
>
> This could be another method to set the termination condition.
>
> *For Figure 6, energy of sampled edges are almost identical for e_optimized and e_allpairs which indicates sampled_edges has a better approximation than embedding_game. Why do you need RMS(gradients) other than RMS(energy) to tell the approximation quality?*
>
> The surprising thing about Figure 6 is that RMS(gradients) is better for embedding game! Intuitively, one might expect that because energy(edge_based_approx) is better than energy(embedding_game), the gradients would also be closer. This intuition turns out to be wrong. Future work should investigate and explain this finding.

---

### Official Review · Reviewer_a79m · 2022-07-12

**Rating:** 2
**Confidence:** 3
**Soundness:** 3 good
**Presentation:** 2 fair
**Contribution:** 2 fair

**Summary:**

This paper proposes a min-max optimization-based dimension reduction technique that approximates the sum O(N^2) using fewer landmarks. They derived a landmark-based lower bound for the repulsion term and then tried to maximize that to approximate the objective. They showed that their optimization requires much fewer landmarks than taking randomly designated points as landmarks.

**Questions:**

1. Could you explicitly elaborate in your paper on why the landmark-based approach should be taken?
2. Have done any exploration on improving the min-max optimization technique for this setup?
3. There is no mention of sample complexity. Do you have any suggestions on what size of L to use based on other factors like, the number of data points, initial dimension size, etc? If yes, please mention them in the paper.


**Limitations:**

Same as the weakness points.

**Strengths And Weaknesses:**

Weakness:
1. The authors have some strong assumptions for their derivations, whereas they did not explicitly mention when would their method fail.
2. There was no analysis of time complexity for their algorithm.
3. Their empirical evaluation, in many cases suggests that sampled edges are better. More empirical evaluation is required.

---

> ### Author Response · Authors · 2022-08-02
> **Reply to a79m**
>
> *The authors have some strong assumptions for their derivations, whereas they did not explicitly mention when would their method fail.*
>
> We are not sure exactly which strong assumption you are referring to. The one key assumption is that g be a kernel function, which is explicitly stated. As far as convergence goes, we do not make any claims that GDA is guaranteed to converge, although provide experiments showing it useful for optimization.
>
> *There was no analysis of time complexity for their algorithm.*
>
> We can add discussion to the paper. Let D = embedding dim, N=# embeddings, L = # landmarks, K=#neighbors. Each iteration of the optimization is NKD + NLD + L^2D.
>
> *Their empirical evaluation, in many cases suggests that sampled edges are better. More empirical evaluation is required.*
>
> Yes, we agree with this statement. However, we believe these experiments are useful for dimensionality reduction researchers nonetheless. Notably, we are unaware of any such explicit comparisons between edge and landmark based methods in the context of nonlinear dimensionality reduction.
>
> *Could you explicitly elaborate in your paper on why the landmark-based approach should be taken?*
>
> Our experiments suggest that the edge-based approach leads to overall higher quality embeddings. Therefore we would still recommend edge-based approaches for practical use. However we believe our experiments are still very useful for researchers in the field of nonlinear dimensionality reduction. Our landmark approach seems like a natural idea, and to our knowledge this is the first paper that actually compares edge based vs landmark based schemes for approximating the all pairs sum. The fact that the edge-based scheme is so much better than randomly sampled landmarks, and still somewhat better than optimized landmarks, is an interesting empirical finding.
>
> *Have done any exploration on improving the min-max optimization technique for this setup?*
>
> To make this method more practical, having a robust optimization method is desired. Unfortunately, we are unaware of any clear methods with strong theoretical guarantees for performing nonconvex-nonconcave optimizations, so this will have to be left for future work.
>
> *There is no mention of sample complexity. Do you have any suggestions on what size of L to use based on other factors like, the number of data points, initial dimension size, etc? If yes, please mention them in the paper.*
>
> Unfortunately, we have no theoretical insights beyond the one already provided in the paper, which is that when L (# landmarks) = N (# data points), the inequality becomes an equality (in other words our bound is in fact equal to the original sum over all pairs)

---

### Official Review · Reviewer_bFeS · 2022-07-18

**Rating:** 3
**Confidence:** 4
**Soundness:** 1 poor
**Presentation:** 3 good
**Contribution:** 2 fair

**Summary:**

The paper addresses the problem of dimension reduction for visualization. It proposes a minimax objective for this problem with game theoretical interpretation. It also proposes a gradient descent-ascent (GDA)-like algorithm to solve this objective that uses non-randomly selected landmarks. They provide empirical results from running their method on the MNIST, scRNA and 25K Reddit Posts datasets.

**Questions:**

Questions on the objective/ formulation.
 - Equation 3 is obtained by dividing Equation 2 by $\sum_{a,b}$, maximizing over $z$ and then plugging the inequality back into Equation 1. But this means that Equation 3 is a lower bound for Equation 1. Does it make sense then to minimize Equation 3 as a proxy for minimizing Equation 1?
 - The section on Duality (lines 147 to 154) is confusing in light of Equation 3. The paper says that there is duality gap in the optimization problem in Equation 3 that means that the ordering of optimization matters. From the von Neumann minimax theory, does this suggest that Equation 3 is not a two-player zero-sum game as stated?

Questions on Algorithm 1.
 - Line 143 says that the optimization problem is nonconvex-nonconcave. Therefore, is it suitable to use GDA as a basis for Algorithm 1 when GDA is known to converge to limit cycles or diverge in this setting? The paper cites Lin et al. (2019) On Gradient Descent Ascent (GDA) for Nonconvex-Concave Minimax Problems which points specifically to this issue. They also mention that it is difficult to set the learning rate so that the algorithm finds a solution. Given these issues, are there other algorithms that could be used instead of GDA?
 - Given the importance of setting the learning rate(s) for Algorithm 1, it could be useful to have this issue addressed in more detail. Lines 118-121 conclude that decreasing the learning rate improves the resulting visualization. It is generally the case, however, that using a smaller stepsize improves accuracy while increasing the iteration complexity of a method. Could the authors provide details such as maximum stepsize allowed for Algorithm 1 not to diverge? In Section 4, it is also unclear what is meant by setting "the landmark learning rate to be relatively fast compared to the embedding learning rate". Could the authors provide details on the relative size of the two learning rates?
 - The paper could be strengthened with more details on the efficiency of the method. For example, could the authors provide theoretical analysis to show the rate of convergence of their algorithm (i.e. a bound on n\_iter in Algorithm 1)? In terms of empirical results, there could be more clarity given too. For example, it is unclear where the plots in Figure 6 came from (e.g. which dataset(s), the learning rate used, etc). Could this be clarified? Could the authors provide similar plots as in Figure 6 but with clock time taken in the x-axes? This is especially useful as the paper argues that one advantage of Algorithm 1 over KMeans is that it provides the landmarks as part of the process. Yet, like running KMeans, there is a cost associated with finding these non-random landmarks too.

Is the code used for the experiments available publicly?

**Limitations:**

This work does not appear to me to have negative social impact.

**Strengths And Weaknesses:**

Strengths
 - The paper addresses an important topic: dimension reduction for visualization.
 - The paper is clearly written with only minor notation issues.
 - The approach of combining optimization and game theory is interesting.
Weaknesses
 - Some questions remain on the validity of the proposed formulation of the problem and objective (Equations 3 and $E$ in equations 4 and 5).
 - More details needed for their proposed method (Algorithm 1).

---

> ### Author Response · Authors · 2022-08-02
> **Reply to bFeS**
>
> *Equation 3 is obtained by dividing Equation 2 by $\sum_{a,b}$, maximizing over z and then plugging the inequality back into Equation 1. But this means that Equation 3 is a lower bound for Equation 1. Does it make sense then to minimize Equation 3 as a proxy for minimizing Equation 1?*
>
> While the more conventional method in machine learning is to minimize upper bounds, rather than lower bounds, giving rise to min_min optimizations instead of min_max optimizations, there is nothing to prevent you from minimizing an upper bound. If the upper bound is accurate enough, this can give rise to useful objectives. We show in the experimental section that performing this min-max gives rise to very nice looking clusterings.
>
> *The section on Duality (lines 147 to 154) is confusing in light of Equation 3. The paper says that there is duality gap in the optimization problem in Equation 3 that means that the ordering of optimization matters. From the von Neumann minimax theory, does this suggest that Equation 3 is not a two-player zero-sum game as stated?*
>
> There appears to be a misunderstanding. Equation 3 is indeed a two player zero sum game. von Neumann minimax theory says that if a two player zero sum objective is convex-concave, then there is no duality gap. This theory is not applicable to our objective as it is not convex concave.
>
> *Line 143 says that the optimization problem is nonconvex-nonconcave. Therefore, is it suitable to use GDA as a basis for Algorithm 1 when GDA is known to converge to limit cycles or diverge in this setting?*
>
> GDA is indeed still applicable. Empirically we show it works quite well. GDA may not be the optimal algorithm, and may in theory have convergence challenges. However, we did find that simply setting the maximizers learning rate to 10x the minimizers learning rate worked well across all experiments.
>
> *Given the importance of setting the learning rate(s) for Algorithm 1, it could be useful to have this issue addressed in more detail. Lines 118-121 conclude that decreasing the learning rate improves the resulting visualization. It is generally the case, however, that using a smaller stepsize improves accuracy while increasing the iteration complexity of a method. Could the authors provide details such as maximum stepsize allowed for Algorithm 1 not to diverge? In Section 4, it is also unclear what is meant by setting "the landmark learning rate to be relatively fast compared to the embedding learning rate". Could the authors provide details on the relative size of the two learning rates?*
>
> We dedicate section 4 to the issue of choosing the learning. Unfortunately, as in most nonconvex optimization problems, providing a maximum allowable step size is nearly impossible. In section 4, we will add a sentence saying that we use landmark learning rate = 10x embedding learning rate in all our experiments.
>
> *The paper could be strengthened with more details on the efficiency of the method. For example, could the authors provide theoretical analysis to show the rate of convergence of their algorithm (i.e. a bound on n_iter in Algorithm 1)? In terms of empirical results, there could be more clarity given too. For example, it is unclear where the plots in Figure 6 came from (e.g. which dataset(s), the learning rate used, etc). Could this be clarified? Could the authors provide similar plots as in Figure 6 but with clock time taken in the x-axes? This is especially useful as the paper argues that one advantage of Algorithm 1 over KMeans is that it provides the landmarks as part of the process. Yet, like running KMeans, there is a cost associated with finding these non-random landmarks too.*
>
> Like any nonconvex optimization, it is nearly impossible to provide theoretical analysis of convergence rates. We can clarify figure 6 for the final paper. (it came from the mnist dataset). We can add a sentence describing the wall clock time for the figures.
>
> *Is the code used for the experiments available publicly?*
>
> We did not release the code.

---

> > ### Comment · Reviewer_bFeS · 2022-08-08
> > **Response after reading rebuttal**
> >
> > Thank you for the replies to my questions and comments. While some of my concerns were addressed, there remains a few points that I am not entirely comfortable with (e.g. minimizing a lower bound of your objective, no theoretical justification for using GDA nor acknowledgement of its flaws in the paper, etc). Thus, I will maintain my rating at 3. Thanks again for your response.

---

### Meta-Review · Area_Chair_ACTA · 2022-08-27

**Recommendation:** Reject
**Confidence:** Certain

**Metareview:**

While the reviewers agreed that the paper addresses an important topic, and combining optimization with game theory is interesting, the reviewers had a number of concerns regarding the validity of the proposed formulation of the problem, lack of theoretical justification for using GDA, strong assumptions, lack of complexity analysis, and limited empirical evaluations. Unfortunately, those concerns were not fully addressed by the author response.

**Award:**

No

---

### Decision · Program_Chairs · 2022-09-14

Reject